# Designing and immunomolecular analysis of a new broad-spectrum multiepitope vaccine against divergent human papillomavirus types

**Maryam Ehsasatvatan**, **Bahram Baghban Kohnehrouz** *

Department of Plant Breeding & Biotechnology, Faculty of Agriculture, University of Tabriz, Tabriz, Iran

* bahramrouz@yahoo.com

**Data Availability Statement:** All relevant data are within the paper and its supporting information files.

## Abstract

Human papillomavirus (HPV), which is transmitted through sexual activity, is the primary cause of cervical cancer and the fourth most common type of cancer in women. In this study, an immunoinformatics approach was employed to predict immunodominant epitopes from a diverse array of antigens with the ultimate objective of designing a potent multiepitope vaccine against multiple HPV types. Immunodominant B cell, cytotoxic T cell (CTL), and helper T cell (HTL) epitopes were predicted using bioinformatics tools These epitopes were subsequently analyzed using various immunoinformatics tools, and those that exhibited high antigenicity, immunogenicity, non-allergenicity, non-toxicity, and excellent conservation were selected. The selected epitopes were linked with appropriate linkers and adjuvants to formulate a broad-spectrum multiepitope vaccine candidate against HPV. The stability of the multiepitope vaccine candidate was confirmed through structural analysis, and docking results indicated a high affinity for Toll-like receptors (TLR2 and TLR4). Molecular dynamics simulations demonstrated a persistent interaction of TLR2 and TLR4 with the multiepitope vaccine candidate. In silico immunological simulations showed that three injections of the multiepitope vaccine candidate resulted in high levels of B- and T-cell immune responses. Moreover, the *in silico* cloning results indicated that the multiepitope vaccine candidate could be expressed in substantial amounts in *E. coli*. The results of this study imply that designing a broad-spectrum vaccine against various HPV types using computational methods is plausible; however, experimental validation and safety testing to confirm the findings is essential.

## Introduction

Human papillomavirus (HPV) is a highly prevalent sexually transmitted infection that affects approximately 75–80% of men and women of all ages [1]. HPV has been linked to the development of several serious cancers and diseases, including cervical and anal cancers, head and neck squamous cell carcinoma (HNSCC), genital warts, and recurrent respiratory

**Funding:** This work is based on research funded by Iran National Science Foundation (INSF) under project No. 4023935. The funders had no role in the study design, data collection and analysis, decision to publish, or manuscript preparation.

**Competing interests:** The authors have declared that no competing interests exist.

**Abbreviations:** HPV, Human papilloma virus; L1, major capsid protein; IEDB, Immune Epitope Data Base web server; MHC-I, Major Histocompatibility Complex class I; MHC-II, Major Histocompatibility Complex class II; HLA, Human leucocyte antigen; pI, Isoelectric point; GRAVY, Grand average of hydropathicity; TLR, Toll like Receptor; PDB, Protein Data Bank; CAI, Codon adaptation index; JCAT, Java Codon Adaptation Tool..

papillomatosis (RRP) [2–4]. To date, more than 200 HPV types have been identified, with some primarily infecting cutaneous tissues and causing warts, whereas others primarily target mucosal tissues of the cervical and oral tracts [4, 5]. Based on their oncogenic potential, HPV types have been grouped into high-risk or oncogenic HPV, which have the potential to cause cancer, for example, HPV16, 18, 31, and 33, or low-risk or non-oncogenic HPV, which are mostly associated with warts, such as HPV6 and 11 [6, 7].

HPVs are small, non-enveloped, double-stranded DNA viruses that infect both the mucosal and cutaneous epithelial cells. The HPV genome contains eight open reading frames (ORFs) organized into three functional regions, including early (E), late (L), and non-coding sections known as long control regions (LCR). E region genes encode proteins E1, E2, E4, E5, E6, E7, and E8, which are essential for viral replication and play a role in the pathogenicity of the virus. The L region genes encode L1 and L2 capsid proteins, which are necessary for virion assembly. Furthermore, LCR genes are vital for the replication and transcription of viral DNA and have tropism for epithelial cells [8, 9].

The L1 protein is the major component of the capsid and comprises both constant and variable regions responsible for surface antigenicity and interaction with host plasma membrane receptors before penetration. Additionally, these regions serve as targets for host antibody generation. Given these characteristics, it is not surprising that L1 plays a key role in the highly variable HPV genotypes. Furthermore, L1 is the target of several ongoing medicines and vaccines because it can bind to high-affinity domains in the infected host and initiate an appropriate immune response as well as form large, non-infectious, but highly immunogenic self-assemblies in the form of virus-like particles (VLPs) [10]. VLPs are known for their ability to trigger a rapid immune response by activating B cells. Once VLPs adhere to the basal membrane, the mature virus binds to the exterior of host cells through the L1 protein. This action causes the L1 protein to become flexible, ultimately releasing the viral genome into the host cell and facilitating viral replication [10].

It has been demonstrated that vaccines that trigger cytotoxic cell activation and cellular immune responses can effectively prevent viral infections. Various vaccines are available for treating viral infections, including prophylactic and therapeutic vaccines [11, 12]. Prophylactic vaccines are designed to activate humoral immunity and promote virus-neutralizing antibody production, thereby preventing viruses from infecting the host cells and providing efficient protection against HPV infection. Currently, three licensed prophylactic vaccines are available to prevent HPV infection: Gardasil, Gardasil-9, and Cervarix. These vaccines were developed using HPV L1 capsid proteins and recombinant DNA technology, which self-assemble into non-infectious virus-like particles (VLPs) and are administered intramuscularly. VLPs do not contain any viral DNA genome or live HPV, making them noninfectious and nononcogenic. Cervarix contains HPV16 and 18 VLPs. Gardasil includes VLPs against HPV6, 11, 16, and 18, whereas Gardasil 9 contains five additional VLPs against HPV31, 33, 45, 52, and 58 [13].

Since 2006, when the HPV vaccine was first approved in the USA, real-world data have consistently demonstrated its safety and effectiveness in preventing and treating HPV infections and its associated diseases. However, various barriers have been identified, such as high vaccine costs, inaccessibility, and inadequate storage and transportation facilities [14]. Moreover, in many low- and middle-income countries, public awareness regarding HPV-related disorders and national vaccination programs is lacking [15]. The main challenge in implementing HPV vaccines is that they do not provide protection against all HPV types [16]. To address these limitations, the next generation of HPV vaccines should focus on reducing the side effects associated with current vaccines using alternative adjuvants or other vaccine designs and high-valent vaccines based on recombinant vectors with a broad protection spectrum that

can be administered by inhalation or the oral route. This is a major step in the treatment of cervical cancer.

Recent studies have successfully used polypeptides that contain multiple epitopes to generate robust immune responses. Similar to long-peptide full/subunit vaccines, any designed polypeptide binds to MHCs via intracellular proteolytic pathways, resulting in more durable epitope presentation and immune responses [17, 18]. To design an effective polypeptide vaccine, it is essential to consider properties, such as intracellular processing, proteasomal cleavage, and MHC binding. The conventional method for creating vaccines is a lengthy and costly process that requires the cultivation of pathogens in a laboratory setting. Immunoinformatics offers a rapid, accurate, and reliable approach for designing vaccines against virulent infections [19].

Our study aimed to design and develop a multiepitope vaccine against the most prevalent low- and high-risk human papillomaviruses, using the major capsid protein L1. To enhance the efficacy of immune responses for an extended period, the designed multiepitope vaccine contained proprietary adjuvants CTB and RS09. MHC-I- and MHC-II-binding epitopes were used as predicted molecules to design both B- and T-cell epitopes, respectively, resulting in a final vaccine composed of multiple epitopes. Molecular dynamics, molecular docking, and in silico expression analyses were performed to evaluate the stability profile and immunological response of the designed multiepitope vaccine.

## Methods

### Retrieving protein sequences and conservancy analysis

The amino acid sequences of the HPV L1 protein for five low-risk HPV types (HPV6, 11, 42, 43, 44) and 12 high-risk HPV types (HPV16, 18, 31, 33, 35, 39, 45, 51, 52, 56, 58, 59) were obtained in FASTA format from the Universal Protein Resource (UniProt) database (http://www.uniprot.org/) in FASTA format (Accession number and amino acid sequences are provided in S1 Table in S1 File). To obtain an effective vaccine against a wide range of HPV types, it is necessary to identify conserved sequences within the viral protein sequences of various strains in order to select efficient candidate epitopes. Clustal Omega was used to perform multiple sequence alignment [20]. After protein alignment analysis, the conserved sequences of the L1 protein were used for immunoinformatic analysis, such as B- and T-cell epitope prediction.

### Prediction of Linear B Lymphocyte (LBL) epitopes

The ABCpred (http://crdd.osdd.net/raghava/abcpred/) [21], BepiPred 3.0 server (https://services.healthtech.dtu.dk/services/BepiPred-3.0/) [22], and Emini Surface Accessibility Prediction tool of IEDB (http://tools.iedb.org/bcell/result/) [23] were used to identify linear B-cell epitopes. In BepiPred 3.0, the epitope threshold value was set at 0.15 (with specificity and sensitivity of 0.57 and 0.58, respectively). ABCpred predicted linear epitopes with an overall accuracy of 65.93% using neural networks. A score closer to one, indicates a higher likelihood of the sequence being an epitope. The Emini method was used to predict surface-accessible epitopes, with a default threshold value of 1.0. The regions predicted by all three tools were selected for further analysis.

The Antigenicity of the selected epitopes was predicted using the VaxiJen v2.0 server (http://www.ddg-pharmfac.net/vaxijen/VaxiJen/VaxiJen.html) [24]. VaxiJen v2.0 predicts immunological scores by transforming protein sequences into uniform vectors of the main amino acid characteristics using auto-cross covariance (ACC). The selection threshold was set at 0.4 and LBL epitopes with immune scores greater than 0.4 were selected for further analysis. Allergen FP v.1.0 (http://ddg-pharmfac.net/AllergenFP/) [25] and AllerTOP v.2.0 (http://ddg-

pharmfac.net/AllerTOP/) [26] were used to predict cellular epitope allergenicity. ToxinPred2 (webs.iiitd.edu.in/raghava/toxinpred2/index.html) [27] was used to predict the epitope toxicity. PyMOL (PyMOL Molecular Graphics System, version 1.2r3pre, Schrödinger, LLC) was used to visualize the precise locations of the selected epitopes inside the L1 protein. Epitopes that were entirely exposed on the surface of the L1 protein were selected because they are readily accessible by antibodies. The degree of variability and conservancy of each epitope was calculated using the IEDB epitope conservancy tools (http://tools.immuneepitope.org/tools/conservancy/) and epitopes with $\geq$ 70% identity were retained.

## Prediction of Cytotoxic T Lymphocyte (CTL) epitopes

The IEDB's major MHC I server (http://tools.iedb.org/mhci/) was used to predict the CTL epitopes. The prediction method was set to IEDB recommended 2023.09 (NetMHCpan 4.1 EL) [28], the MHC source species to Human, the MHC allele(s) to the human leukocyte antigen (HLA) allele reference set, and the epitope length to 9 and 10. Peptides with a percentile rank less than 0.5 were chosen for further analysis. The antigenicity of the epitopes was predicted using VaxiJen v2.0, with a threshold set at 0.4, and epitopes with a score greater than 0.4 were selected for further exploration. Subsequently, the Class I Immunogenicity Server (http://tools.iedb.org/immunogenicity/) [29] was used to analyze the immunogenicity of these epitopes. Epitopes with positive scores were selected for further investigation. Cellular epitope allergenicity was predicted using AllerTOP v.2.0 and Allergen FP v.1.0, and ToxinPred predicted epitope toxicity. Finally, epitopes that passed the screening were selected as immunodominant CTL epitopes. The conservation of each epitope was calculated using the IEDB epitope conservancy tool and epitopes with $\geq$ 70% identity were maintained.

## Prediction of Helper T Lymphocyte (HTL) epitopes

The NetMHCIIPan 4.0 server (http://www.cbs.dtu.dk/services/NetMHCIIpan/) [30] and the consensus method of the IEDB MHC-II prediction tool (http://tools.immuneepitope.org/mhcii) [31] were used to identify peptides that bind MHC-II molecules. The prediction method in IEDB was set to the IEDB approved 2023.05 (NetMHCIIPan 4.1 EL), with Human as the MHC source species, the 7-allele HLA reference set as the MHC allele(s), and length as 12–18. The epitopes with lower percentile rank indicate higher binding affinity; thus, epitopes with less than 1.0 percentile rank were selected for further exploration. The NetMHCIIpan 4.0 server, using an artificial neural network algorithm, classifies sequences as strong binding peptides (SB) or weak binding peptides (WB) based on %Rank greater than 1.0, or 5, respectively. Antigenicity of the epitopes was predicted using VaxiJen v2.0, with a threshold of 0.5. Epitopes with a score less than 0.5 were selected for further analysis. Cellular epitope allergenicity was predicted using AllerTOP v.2.0 and Allergen FP v.1.0, and ToxinPred predicted epitope toxicity. Finally, the IFN-γ Epitope Server (webs.iiitd.edu.in/raghava/ifnepitope/index.php) was utilized to forecast the inducibility of epitopes for interferon-gamma (IFN-γ), with the model set as IFN-γ versus non-IFN-γ and the method set as Motif and SVM hybrid. Immunodominant HTL epitopes were selected among epitopes that passed screening. The conservation of each epitope was calculated using the IEDB epitope conservancy tool, and epitopes with $\leq$ 70% identity were retained.

## Population coverage analysis of the selected epitopes

The selected CTL and HTL epitopes and their corresponding HLA alleles were subjected to population coverage rates analysis using the IEDB population coverage tool (http://tools.iedb.

org/population/) [32] The analysis was conducted on the entire global population with the default parameters retained. The evaluation included both HLA classes I and II.

## Multiepitope vaccine construct design

A multiepitope vaccine candidate was designed by connecting conserved CTL, HTL, and B-cell linear epitopes from the most prevalent low- and high-risk HPV types, using linkers to prevent HPV. B-cell epitopes were linked together using the KK linker as it has been shown to preserve the immune activity of vaccine epitopes and stabilize the protein structure [33, 34]. AAY and GPGPG linkers were used to connect immunodominant CTL and HTL epitopes, respectively, because of their advantages in terms of length, flexibility, and capacity to speed up immune processing and presentation [35–37]. Furthermore, cholera toxin subunit B (CTB) (TPQNITDLCAEYHNTQIYTLNDKIFSYTESLAGKREMAIITFKNGAIFQVEVPGSQHIDS QKKAIERMKDTLRIAYLTEAKVEKLCVWNNKTPHAIAAISMAN) [38] and TLR4 agonist RS09 (sequence: APPHALS) [39, 40] were added as adjuvants to the C- and N-terminus of the multiepitope vaccine candidate, respectively, based on their capacity to improve antigen pre-sentation and immune responses. A 6×His-tag (HHHHHH) was incorporated into the N-ter-minus of the vaccine candidate to facilitate protein detection.

## Antigenicity, allergenicity, toxicity and solubility prediction of multiepitope vaccine candidates

ANTIGENpro (http://scratch.proteomics.ics.uci.edu/) and VaxiJen 2.0 (with a threshold set at 0.5) were used to evaluate the antigenicity of the multiepitope vaccine candidate. ANTIGEN-pro is a sequence-based, alignment-free, pathogen-independent predictor of whole-protein antigenicity. It was trained on reactivity data from a protein microarray analysis of five patho-gens and its experimental accuracy on the combined dataset was 76% [41]. Allergen FP v. 1.0 and AlllerTOP v. 2.0, were used to analyze the allergenicity of the vaccine candidate because identifying allergens is crucial for vaccine development. Allergen FP v.1.0 has an accuracy of 88% in identifying known allergens and non-allergens [25], whereas AlllerTOP v.2.0 has an accuracy of 85.3% at five cross-validations [26]. The toxicity of the multiepitope vaccine candi-date was predicted using the ToxinPred2 server with Random Fores based on amino acid com-position and a threshold of 0.5. Finally, the Protein-Sol server (https://protein-sol.manchester.ac.uk/) [42] was used to forecast the solubility of the vaccine candidate. This server uses a sin-gle amino acid sequence to perform a series of solubility prediction computations and provides the results, which are subsequently compared to a solubility database. According to the experi-mental solubility dataset, a prediction value more than 0.45 implies that the protein has better solubility than the average *E. coli* soluble protein.

## Cross-reactivity with host antigens

Adverse immune responses may arise because of cross-reactivity with host antigens. Conse-quently, it is crucial to verify the selected peptides at the initial stage for similarities with the proteomes of mice and humans. The entire multi-epitope vaccine sequence, as well as its indi-vidual epitopes, were submitted for analysis using the NCBI BLASTp tool (https://blast.ncbi.nlm.nih.gov/Blast.cgi).

## Physiochemical properties estimation and secondary structure prediction

The Expasy Protparam server (https://web.expasy.org/protparam/) [43] was used to predict the physicochemical properties of the multiepitope vaccine candidate, including its molecular

weight, amino acid composition, atomic composition, theoretical isoelectric point (pI), instability index, extinction coefficient, aliphatic index, estimated half-life, and GRAVY (grand average of hydropathicity). The Prabi servers (https://npsa-prabi.ibcp.fr/cgi-bin/npsa_automat.pl?page=/NPSA/npsa_gor4.html) [44] was used to predict the secondary structure of the multiepitope vaccine candidate. PSIPRED (http://bioinf.cs.ucl.ac.uk/psipred/) [45] was utilized to predict the transmembrane helix, transmembrane topology, folds, and domain recognition of multiepitope vaccine candidate.

## Prediction, optimization, and quality assurance of tertiary structure

AlphaFold2 [46], a cutting-edge machine learning technique for protein structure prediction that covers nearly the whole human proteome (98.5% of human proteins), was utilized to predict the 3D model of the multiepitope vaccine candidate. The highest rank predicted model was refined using the GalaxyRefine web server (https://galaxy.seoklab.org/cgi-bin/submit.cgi?type=REFINE) [47] to enhance accuracy. ERRAT [48] and Procheck Ramachandran Plot v.3.5.4 [49] of the SAVES V6.0 web tool and ProSA-Web [50] were used for structural evaluation and stereochemical analyses. ERRAT evaluates the non-bonded interaction statistics between different types of atoms in the predicted structure and a higher score on the server indicates a higher-quality model. The Ramachandran plot is an essential tool for evaluating protein model quality and validity, providing a visual representation of residue distribution in favored, allowed, and outlier regions. In addition, the ProSA Z-score offers a comprehensive model quality evaluation by estimating the total energy of the structure based on the energy distribution produced by random conformations.

## Prediction of conformational B-cell epitopes

The majority of B-cell epitopes are conformational in nature, comprising one-five linear fragments of amino acid residues. These conformational B-cell epitopes are discontinuous and located at distant positions in the sequence; however, they are in close spatial proximity and serve as sites for antibody interactions [51]. The multiepitope vaccine intends to fold differently, forming conformational B cell epitopes that induce an adaptive immune response [52]. Therefore, the accurate prediction of discontinuous B-cell epitopes is crucial for refining the structure of multiepitope vaccines in space. The ElliPro server (http://tools.iedb.org/ellipro/) [53], a widely used nonlinear B-cell epitope prediction tool, was used to predict discontinuous epitopes. This server is known to achieve the highest level of accuracy for calculating B cell epitopes for any protein. Models with a score greater than 0.5 are typically considered acceptable for predicted conformational B cell epitopes.

## Molecular docking

Molecular docking analysis was conducted using ClusPro 2.0, an online server [54] to simulate the interaction between the multiepitope vaccine candidate and TLR2 and TLR4. An effective immune response induced when a vaccine candidate has a high binding affinity for target immune cells. In molecular docking analysis, the final refined tertiary structure of the designed multiepitope vaccine candidate was submitted as a ligand. The PDB files for TLR2 (PDB ID: 5D3I) and TLR4 (PDB ID: 4G8A) used in docking analysis as receptors were obtained from the RCSB database (http://www.rcsb.org/). The ligands and heteroatoms attached to the retrieved TLR2 and TLR4 structures and chains B, C, and D of TLR4 were removed prior to docking using PyMOL. The interaction between the multiepitope vaccine candidate and TLR2 and TLR4 was evaluated using the ClusPro server. The ClusPro server performs rigid body docking on two proteins using billions of conformations. Docking was performed without the

use of predetermined active site information, and the centers of the largest clusters were used as plausible complex models of low-energy docked structures. The potential binding affinities of the vaccine candidates for TLR2 and TLR4 were determined using the PRODIGY server [55]. PDBsum [56] and using LigPlot + v.2.2 [57] were used to predict critical vaccine-receptor interactions, including salt bridges and hydrogen bonds, and graphical representations were constructed using PyMOL version 2.3.2\_81 (PyMOL Molecular Graphics System).

## Molecular dynamics simulation

The molecular dynamics simulation was used to acquire a thorough knowledge of the dynamics and stability of the docked vaccine-TLR2 and vaccine-TLR4 complexes using GROMACS (version 2019.2) [58], Charmm27 all-atom force field, and well-tested simple point charge (SPC) model for water molecules. Each simulation involved a 50 ns trajectory of the vaccine-receptor complex. The systems were solvated in a triclinic box with a 1.0 nm distance from each wall and periodic boundary conditions and neutralized by the addition of Na+ and Cl- ions. Subsequently, energy minimization was conducted using 5000 steps of the steepest descent minimization algorithm. After then, the system was kept at 300 K for 100 ps in the NVT ensemble and then equilibrated for 100 ps in the NPT ensemble. Lastly, an ensemble production simulation of 50 ns NPT was run. The Particle Mesh Ewald method was utilized to calculate long-range electrostatic interactions, using a collision frequency of 2 ps and a nonbonded cutoff of 1 nm. Trajectory data was saved every 10 ps, with the system pressure maintained at 1 bar and the integration time step set to 2 fs. Trajectory processing was performed using GROMACS tools, which allowed for the computation of several structural parameters, including the Root Mean Square Deviation (RMSD), Root Mean Square Fluctuation (RMSF), Radius of Gyration (Rg), Solvent-Accessible Surface Area (SASA), and hydrogen bond interactions. To ensure the accuracy of the results, three independent simulations were performed for each complex with each simulation run for 50 ns. The binding free energies of the vaccine and TLRs complexes were calculated using molecular mechanism (MM) energies combined with Poisson–Boltzmann (PB) and Surface Area (SA) solvation methods (MM-PBSA) [59]. The last 10 ns of the trajectory were extracted and submitted to the gmx_MMPBSA v1.56 tool [60]. In this study, the MM-PBSA method was adopted to predict the binding free energy of the complex, as follows:

$$\Delta G_{bind} = \Delta G_{complex} - (\Delta G_{ligand} + \Delta G_{receptor})$$

## *In silico* immune response simulation

The immunogenic properties of the designed multiepitope vaccine candidate were simulated utilizing the C-ImmSim server (http://150.146.2.1/CIMMSIM/index.php), which is an agent-based computer model that uses a position-specific scoring matrix (PSSM) to predict immune epitopes and immune interactions. This server predicts the immune response of B and T lymphocytes following a simulated vaccine injection by integrating the immune system into three chambers: the bone marrow, thymus, and lymphatic organs. The server was utilized with the default parameters, a simulation volume of 50, and 1000 simulation steps, with three injections of the predicted vaccine construct administered at intervals of 4 weeks. Each time step in the server was equivalent to 8 h in real life, and the time periods were set to 1, 84, and 168 h. Finally, the cellular immune response and cytokine levels induced by the multiepitope vaccine candidate were predicted.

## Codon optimization and in-silico CLoning

Codon optimization is often necessary to optimize the host-based expression of recombinant proteins, because codons play a crucial role in translating gene information into protein sequence data. Java Codon Adaptation Tool (JCat) (http://www.jcat.de/) was used for reverse translation and codon optimization of the multiepitope vaccine construct for expression in *E. coli* (strain K12). The codon adaptive index (CAI) value and GC content of the optimized sequence were generated using this tool. The ideal CAI value is 1 and the GC content should be between 30% and 70%. We inserted the multiepitope vaccine candidate sequence between the *Nco*I and *Xho*I restriction sites of pET28a(+) plasmids using SnapGene software (version 6.0.2).

# Results

## Sequence alignment

The L1 protein sequences of five low-risk (HPV6, 11, 42, 43, and 44) and 12 high-risk (HPV16, 18, 31, 33, 35, 39, 45, 51, 52, 56, 58, and 59) human papillomaviruses are presented in the supplementary file (S1 Table in S1 File). Multiple sequence alignments of the L1 sequences were performed using the EMBL-EBI Clustal Omega server to select for conserved epitopes between HPV types. Based on the sequence homology of conserved regions, the L1 proteins from one low-risk HPV type (HPV6) and two main high-risk HPV types (HPV16 and 18) were selected as a reference for the prediction of conserved B- and T-cell epitopes (S1 Fig in S1 File).

## Prediction of immunodominant B-cell linear epitopes

Continuous B-cell epitopes were predicted using BepiPred, ABCpred, and the Emini surface accessibility prediction tool on the IEDB servers. Antigenicity analysis was performed with the VaxiJen 2.0, and peptides with an antigen score greater than 0.4 were selected. Following visual analysis, 25 potential epitopes situated entirely on the surface of the L1 protein in both low- and high-risk HPV types were identified. The selected epitopes demonstrated high sequence conservation among both low- and high-risk HPV types, and none were classified as allergens or toxins (S2 Table in S1 File). Furthermore, the peptide showed no cross-reactivity with the mouse or human proteomes.

## Prediction of immunodominant CTL and HTL epitopes

The Immune Epitope Database (IEDB) web server was used to predict the binding of epitopes from L1 protein to MHC class I and II alleles, resulting in the identification of CTL and HTL epitopes, respectively. Antigenicity of each epitope was evaluated using VaxiJen 2.0. HTL epitopes with antigenicity scores greater than 0.5 were evaluated for the interferon-γ (IFN-γ) score and epitopes with positive IFN-γ response were selected, while CTL epitopes with antigenicity scores greater than 0.4 and immunogenicity score greater than 0 were selected. It is worth noting that only non-toxic and non-allergenic epitopes were chosen, and those that produced conflicting results upon evaluation by AllergenFP 1.0 and AllerTop 2.0, only epitopes with consistent non-allergenic predictions were retained. In addition, epitopes containing transmembrane regions or signal peptides were not considered. Based on allelic interactions and high population coverage, 19 CTL and 12 HTL conserved antigenic, nonallergenic, and nontoxic epitopes were selected for further analyses (S3 and S4 Tables in S1 File). Furthermore, the peptide showed no cross-reactivity with the mouse or human proteomes.

**Table 1. The selected B-cell linear epitopes of different types of HPV major capsid protein L1.**

| Epitope | Start position | Length | Antigenicity |
|---|---|---|---|
| VSGYQYRVFKVVLPD | 62 in HPV6 | 15 | 0.9381 |
| TVYLPPVPVSKVVS | 10 in HPV16 | 14 | 0.6496 |
| GDMVDTGYGA | 259 in HPV18 | 10 | 1.3932 |

## Construction of the multiepitope vaccine construct

The selected B-cell linear (Table 1), CTL, and HTL epitopes (Table 2) of the L1 protein from low- and high-risk HPV types were linked to create a multiepitope vaccine candidate using various linkers. Specifically, B-cell linear epitopes were connected using KK linkers, whereas AAY and GPGPG linkers were used to connect CTL and HTL epitopes, respectively. Furthermore, an EAAKK linker was utilized to add adjuvants to the C- and N-terminal of the multiepitope vaccine. The sequence and construction strategy of the multiepitope vaccine candidate are depicted in Fig 1A and 1B.

## Prediction of population coverage of the vaccine candidate

The assessment of the global population coverage of selected CTL and HTL epitopes and concurrent evaluation of their corresponding HLA alleles across diverse ethnicities and geographic regions showed 99.26% coverage for both classes combined, with an average hit of 8.53 (Fig 2). These findings suggest that multiepitope vaccine candidates exhibit extensive recognition potential and widespread applicability, and encompass a substantial number of individuals worldwide.

## Antigenicity, allergenicity and toxicity of the multiepitope vaccine candidates

Various tools have been employed to assess the allergenicity and antigenicity of multiepitope vaccine candidate. According to the results of VaxiJen 2.0 and ANTIGENpro, the antigenicity of the designed multiepitope vaccine was 0.6484 and 0.9436, respectively. Furthermore, Allergen FP v.1.0 and AllerTop v.2.0 revealed that the vaccine candidate was non-allergenic. Moreover, ToxinPred2 indicated that the vaccine candidate was non-toxic (Table 3). These findings imply that multiepitope vaccine candidate may be promising options for vaccination against HPV infections. The result of the search for similarity between the multi-epitope vaccine and Homo sapiens proteins was negative. Additionally, when chimeric epitopes were examined

**Table 2. The selected HTL and CTL epitopes of different types of HPV major capsid protein L1.**

| CTL epitopes | | | | | |
|---|---|---|---|---|---|
| Epitope | Start position | Length | Top Alleles | Antigenicity | Immunogenicity |
| DTGFGAMNF | 198 in HPV6 | 9 | HLA-A*26:01 | 2.1472 | 0.0466 |
| KFGFPDTSFY | 82 in HPV16 | 10 | HLA-A*30:02 | 1.1960 | 0.1049 |
| HVEEYDLQF | 428 in HPV18 | 9 | HLA-A*32:01 | 1.4499 | 0.07349 |
| HTL epitopes | | | | | |
| Epitope | | Length | Alleles | Antigenicity | IFN-γ |
| KFLLQSGYRGRSSIR | 462 in HPV6 | 15 | HLA-DRB5*01:01 | 1.0609 | 0.2297 |
| PSEATVYLPPVPVSK | 6 in HPV16 | 15 | HLA-DRB1*07:01 | 0.5687 | 0.0812 |
| HSMNSSILEDWNFGVPPP | 454 in HPV18 | 18 | HLA-DRB3*01:01 | 0.6607 | 0.1183 |

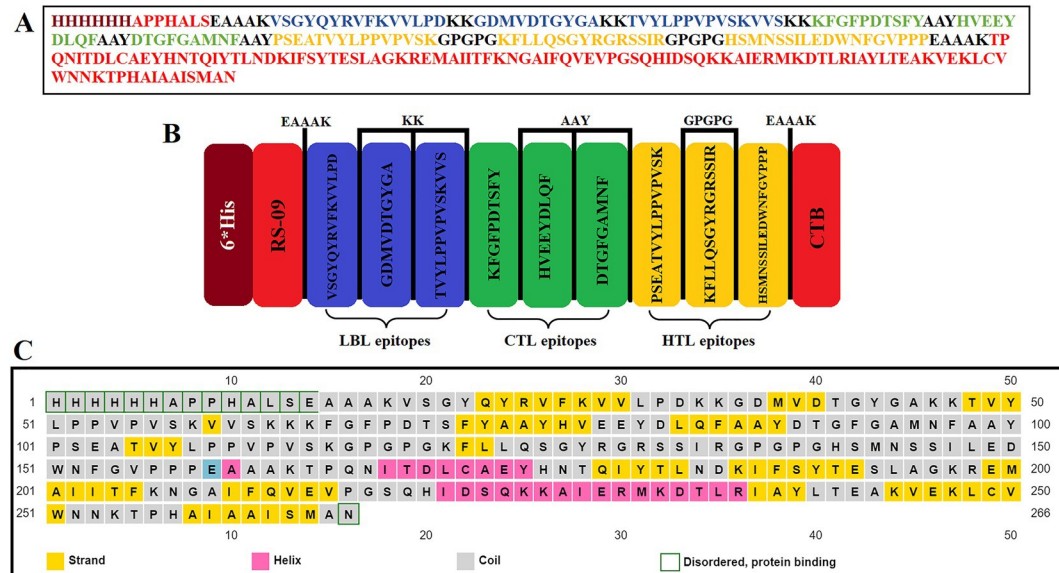

**Fig 1.** A) The amino acid sequences of multiepitope vaccine candidate with different elements exhibited by different colors, B) Structural arrangement of the final vaccine candidate constructed from adjuvant, LBL, CTL, and HTL epitopes separated by linkers; C) The strand, helix, and coil secondary structures in the amino acid sequences of vaccine candidate are shown in yellow, pink, and grey, respectively.

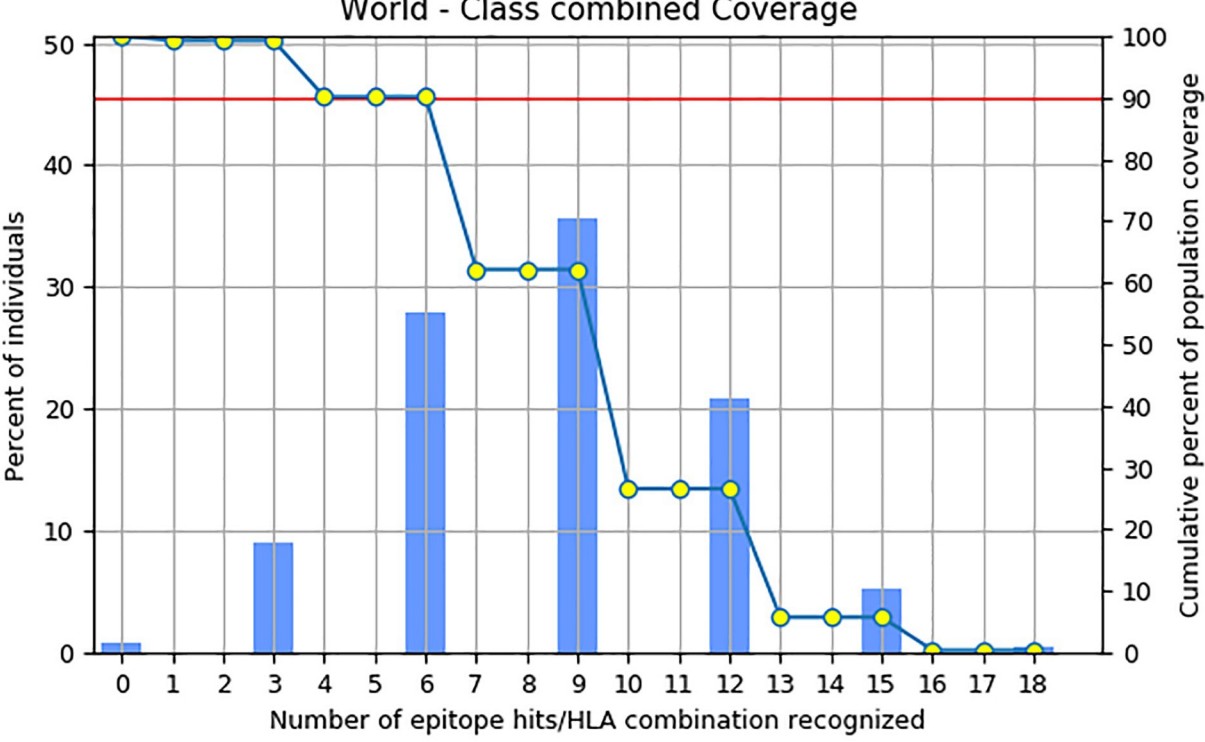

**Fig 2. World population coverage for combined MHC-I and MHC-II alleles showed that the selected epitopes provide coverage for 99.26% of the global population, with an average hit of 8.53.** The bar represents the population coverage for each epitope, whereas the cumulative percentage of population coverage is represented by a line graph (-o-).

**Table 3. Parameters prediction of the multiepitope vaccine candidate.**

| Parameter | Antigenicity | Allergenicity | Toxicity | Solubility | GRAVY Score | Number of aa | Molecular weight (Da) | Isoelectric point | Aliphatic index | Half-life |
|---|---|---|---|---|---|---|---|---|---|---|
| **Result** | 0.6484[a] 0.9436[b] | Non-allergen | Non-toxin | 0.414 | -0.367 | 266 | 29302.38 | 8.96 | 70.83 | 3.5 h[c] 10 min[d] >10 h[e] |

[a] The antigenicity predicted by VaxiJen 2.0

[b] The antigenicity predicted by ANTIGENpro

[c] Mammalian reticulocytes, in vitro

[d] Yeast, in vivo

[e] *Escherichia coli*, in vivo

individually using BLAST, no matches were found. These results suggested that our hypothetical antigen was not self-reactive and might be safe for in vivo assays.

## Physicochemical properties and secondary structure prediction of the multiepitope vaccine candidate

The molecular weight of the multiepitope vaccine candidate was determined to be 29.3 kDa, with 266 amino acids and an isoelectric point of 8.96. In addition, the vaccine candidate GRAVY score was -0.367, with an instability index of 39.29. The aliphatic index for the vaccine candidate was found to be 70.83. Furthermore, the vaccine candidate had a half-life of 3.5 hours in human reticulocytes (in vitro), 10 min in yeast, and > 10 h in E. coli (in vitro) (Table 3). The solubility of the multiepitope vaccine candidate was 4.14, indicating that it was soluble upon expression (S2 Fig in S1 File). The secondary structure of the multiepitope vaccine candidate was analysed using the Prabi and PSIPRED servers, revealing 26.32% alpha helices (70/266), 19.92% extended strands (53/266), and 53.76% random coils (143/266) (Fig 1C).

## Prediction and optimization of tertiary structure of the multiepitope vaccine candidate

The tertiary structure of the multiepitope vaccine candidate was predicted with AlphaFold2. Five models were generated and the top-ranked model was selected for further optimization. The GalaxyRefine web server was used to improve the prediction results, refine the loops, and minimize the energy of the model. Five models were obtained and evaluated using key parameters including RMSD, GDT-HA, and MolProbity scores to account for weak rotamers, clash scores, and Rama-favored regions. More GDT-HA values correspond to more accurate backbone structures, while lower MolProbity scores correspond to better-quality models. The final 3D representation of the multiepitope vaccine candidate was selected as the model with the highest GDT-HA value and smallest MolProbity score among the five optimized models (Fig 3A).

PROCHECK, ERRAT, and ProSA-web servers were used to evaluate and verify the quality and potential errors of the refined 3D model. The Z-score for the multiepitope vaccine candidate was -4.34 (Fig 3B), and the ERRAT overall quality factor was 92.37 (S3 Fig in S1 File). Ramachandran analysis using the PROCHECK server revealed that 94.6% of the residues were in the most favorable regions, 4.9% were in additional allowed regions, 0.4% were in generously allowed regions, and none were in disallowed regions (Fig 3C).

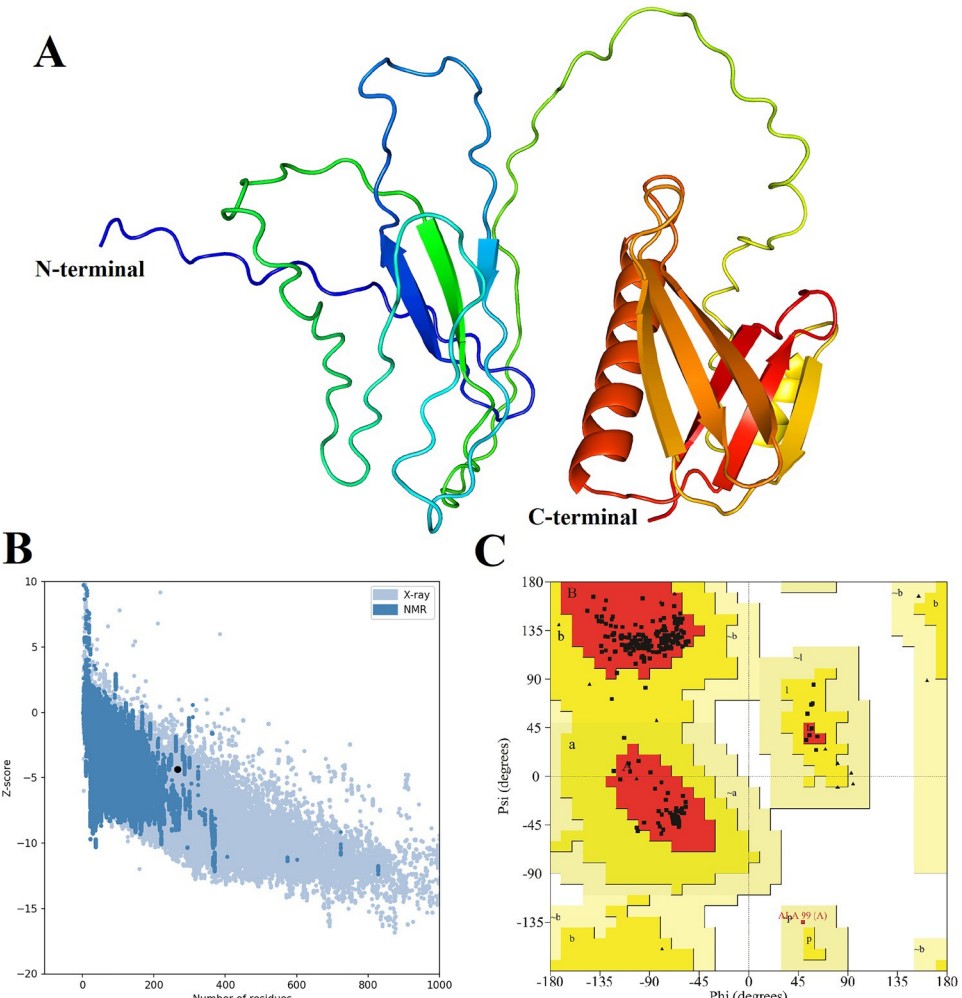

**Fig 3. Validation of the tertiary structure of the multiepitope vaccine candidate.** A) The 3D model structure of the multiepitope vaccine candidate predicted using AlphaFold2 and refined using GalaxyRefine. B) Z-score graphs generated by the ProSA-web server of the refined models of multiepitope vaccine candidate. C) The Ramachandran Plots of multiepitope vaccine candidate show the allowed regions (depicted in orange and deep yellow), the generously allowed regions (in light yellow), the outlier regions (in white), and the glycine residues represented as triangles.

## Conformational B cell epitopes

According to the results from the ElliPro server, 137 residues of the multiepitope vaccine candidate were distributed among 12 conformational B-cell epitopes, scoring between 0.842 and 0.543 (Fig 4 and S4 Fig in S1 File). The epitopes ranged in size from 5 to 17 amino acid residues. Table 4 lists the conformational epitope details and corresponding scores for the multiepitope vaccine candidate.

## Molecular docking between the multiepitope vaccine candidate and TLR2 and TLR4

Antigen-immune receptor interactions are critical for the transport of antigenic molecules and the activation of the immune system. The ClusPro 2.0 server was used for molecular docking to assess the molecular interactions and intensity of binding energy between multiepitope vaccine candidate and immunological receptors (TLR2 and TLR4). Out of 30 possible docked complex configurations, the best

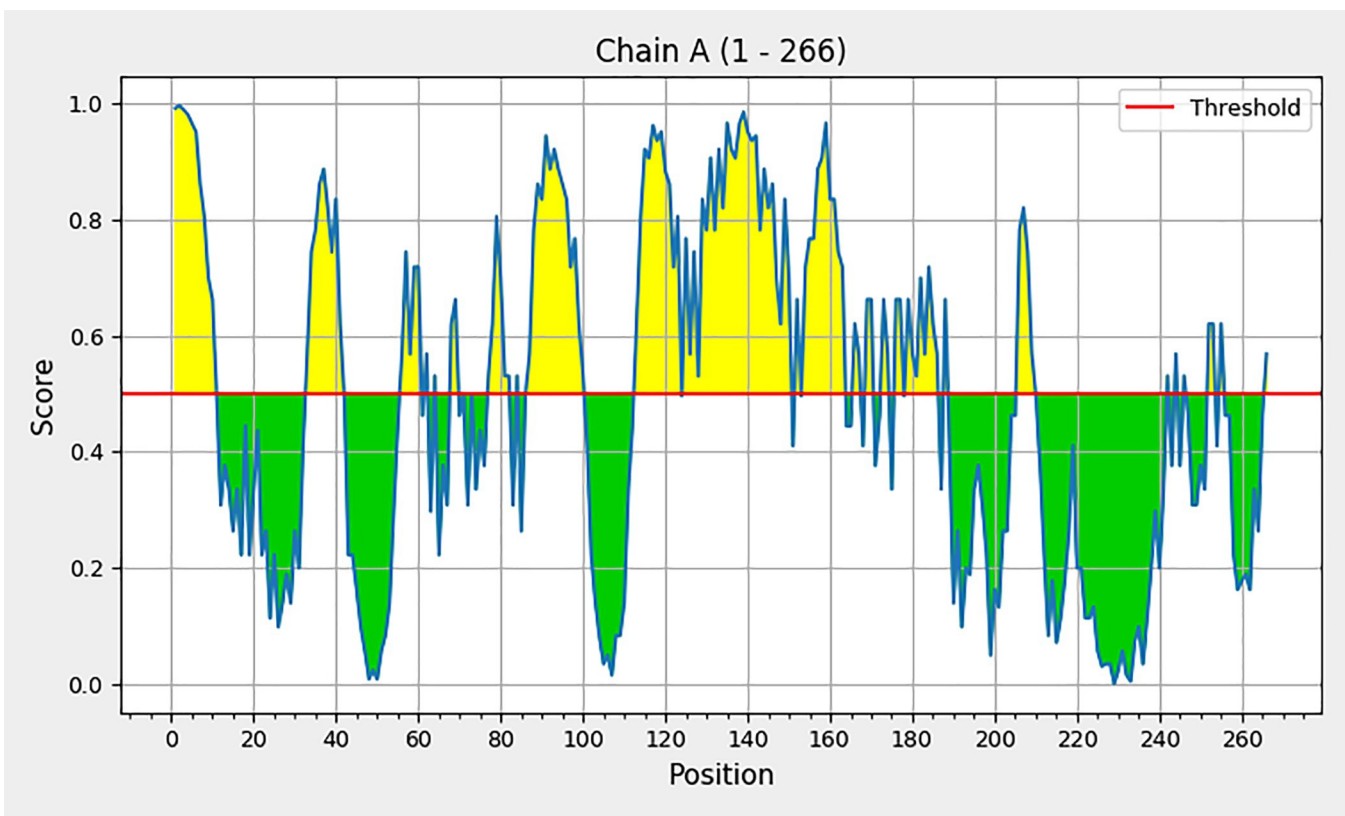

**Fig 4. Discontinuous B cell epitopes predicted by the ElliPro server.** The epitopes are depicted as a graph, where the x-axis represents the amino acid position, and the y-axis represents the score. The yellow region indicates potential B-cell epitopes, with scores exceeding the threshold of 0.5.

Table 4. list of discontinuous B-cell epitopes predicted by the ElliPro server.

| No | Discontinuous B-cell epitope | Number of residues | Score |
|---|---|---|---|
| 1 | A:G130, A:R131, A:S132, A:S133, A:I134 | 5 | 0.842 |
| 2 | A:R135, A:G136, A:P137, A:G138, A:P139, A:G140, A:H141, A:S142, A:M143, A:N144, A:S145, A:S146, A:I147, A:L148, A:E149, A:D150, A:W151 | 17 | 0.835 |
| 3 | A:H3, A:H4, A:H5, A:H6, A:A7, A:P8, A:P9, A:H10, A:A11 | 9 | 0.823 |
| 4 | A:N152, A:F153, A:G154, A:V155, A:P156, A:P157, A:P158, A:E159, A:A160, A:A161, A:A162, A:K163, A:T164 | 13 | 0.75 |
| 5 | A:K18, A:P112, A:V113, A:S114, A:K115, A:G116, A:P117, A:G118, A:P119, A:G120, A:K121, A:F122, A:L123, A:L124, A:Q125, A:S126, A:G127, A:Y128, A:R129 | 19 | 0.747 |
| 6 | A:A86, A:A87, A:Y88, A:D89, A:T90, A:G91, A:F92, A:G93, A:A94, A:M95, A:N96, A:F97, A:A98, A:A99, A:Y100, A:P101 | 16 | 0.745 |
| 7 | A:F188, A:K206, A:N207, A:G208, A:A209 | 5 | 0.715 |
| 8 | A:P32, A:D33, A:K34, A:K35, A:G36, A:D37, A:M38, A:V39, A:D40, A:T41, A:G42 | 11 | 0.706 |
| 9 | A:P165, A:N167, A:T169, A:D170 | 4 | 0.584 |
| 10 | A:H176, A:N177, A:Q179, A:I180, A:Y181, A:N252, A:N253, A:K254, A:T255, A:P256 | 10 | 0.582 |
| 11 | A:P55, A:V56, A:S57, A:K58, A:V59, A:V60, A:S61, A:A75, A:Y76, A:H77, A:V78, A:E79, A:E80, A:Y81, A:D82, A:L83, A:Q84 | 17 | 0.564 |
| 12 | A:T182, A:L183, A:N184, A:D185, A:K186, A:T241, A:E242, A:A243, A:K244, A:E246, A:A265 | 11 | 0.543 |

one was selected based on the centers of highly populated clusters of low energy structures, demonstrating that the vaccine construct effectively binds to the receptor-binding domain. These complexes exhibit strong molecular interactions, including non-bonded contacts, hydrogen bonds, and salt bridges. Between the vaccine candidates and TLR2, we identified 2 salt bridges, 9 hydrogen bonds, and 233 non-bonded interactions. The hydrogen bond-interacting residues in vaccine-TLR2 complex were ASN96-ASP294, HIS10-ILE319, HIS10-LYS147, TYR81-GLU375, GLN23-ARG400, GLU14-ARG400, and LYS18-HIS426. The salt bridge interactions in vaccine-TLR2 complex involved ARG25-GLU375 and GLU14-ARG400. Between the vaccine candidate and TLR4, we found 27 hydrogen bonds, 8 salt bridges, and 230 non-bonded interactions. The hydrogen bonds interacting residues in vaccine-TLR4 complex were: THR90-GLN39, GLU79-ARG264, GLU79-ARG264, GLU79-ASN339, GLU79-LYS362, GLU79-LYS362, ASP82-ARG382, GLU14-HIS431, LEU12-HIS431, HIS4-GLU474, HIS5-GLU474, HIS1-GLU474, PRO8-LYS477, PRO9-LYS477, HIS2-ASN497, HIS1-ASN497, HIS4-THR499, SER13-GLN505, TYR128-ASN531, SER132-ASP580, TYR100-GLU603, MET38-ARG606, THR41-GLU608, GLY42-GLU608, ARG135-SER613, ARG135-SER613, ARG135-ASP614. The salt bridge interacting residues in vaccine-TLR4 complex were: GLU79-ARG264, GLU79-LYS362, ASP82-ARG382, GLU14-HIS431, HIS1-GLU474, HIS5-GLU474, HIS10-ASP502, and ARG135-ASP614. The detailed atom-atom interactions between the multiepitope vaccine candidate and the TLR2 and TLR4 interfaces are shown in Fig 5.

The Gibbs free energy (ΔG) and the dissociation constant (KD) values were obtained from the PRODIGY server to confirm the binding affinity of the multiepitope vaccine candidate for TLRs. The results showed that the vaccine candidate had a higher binding affinity to TLR4 (ΔG = -19.6 kcal.mol$^{-1}$ and KD = 4.5e-15) compared to TLR2 (ΔG = -16.2 kcal.mol$^{-1}$ and KD = 1.4e-12). The number of hydrogen bonds and salt bridges established between the vaccine candidate and TLR4 was higher than that with TLR2, supporting the highest potential for binding of the designed vaccine candidate to TLR4. Molecular docking suggested effective binding and significant affinity between the designed vaccine construct and the immune receptors.

## Molecular dynamics simulation

Molecular dynamics simulations were performed to investigate the stability of the vaccine and TLR2 and TLR4 complexes. Three independent simulations for each complex, each lasting 50 ns, were conducted using the GROMACS version 2019.2. Root-mean-square deviation (RMSD) is an important statistic for determining whether a complex is in equilibrium. For the vaccine-TLR4 complex, RMSD showed that the system reached a constant value after approximately 5 ns and exhibited highly stable behavior for the remainder of the simulation time, with no significant deviations observed, and had an average RMSD of 0.51 nm. The vaccine-TLR2 complex had the highest RMSD value and the least stable structure, as evidenced by the severe fluctuations in its plot. Its RMSD curve reached equilibrium about 37 ns after the start of the simulation, with an average RMSD of 0.77 nm (Fig 6A). To gain deeper insight into the mobility of contact residues and their influence on the binding of the construct to the receptor, the flexibility of the residues in the vaccine-TLRs complexes were analyzed using Root Mean Square Fluctuation (RMSF). The vaccine-TLR2 and vaccine-TLR4 complexes showed average RMSF values of 3.56 Å and 2.31 Å, respectively. Multiepitope vaccine exhibited greater fluctuations. A comparison of the two plots indicated that the complex with TLR4 showed fewer fluctuations, indicating better binding between interacting partners or stable complexes (Fig 6B). The Rg profile was used to assess the regular packing and structural compactness of the secondary structural components in the three-dimensional structure of the complex (Fig 6C). The less stable folding of secondary structural components is linked to loose packing of the structure,

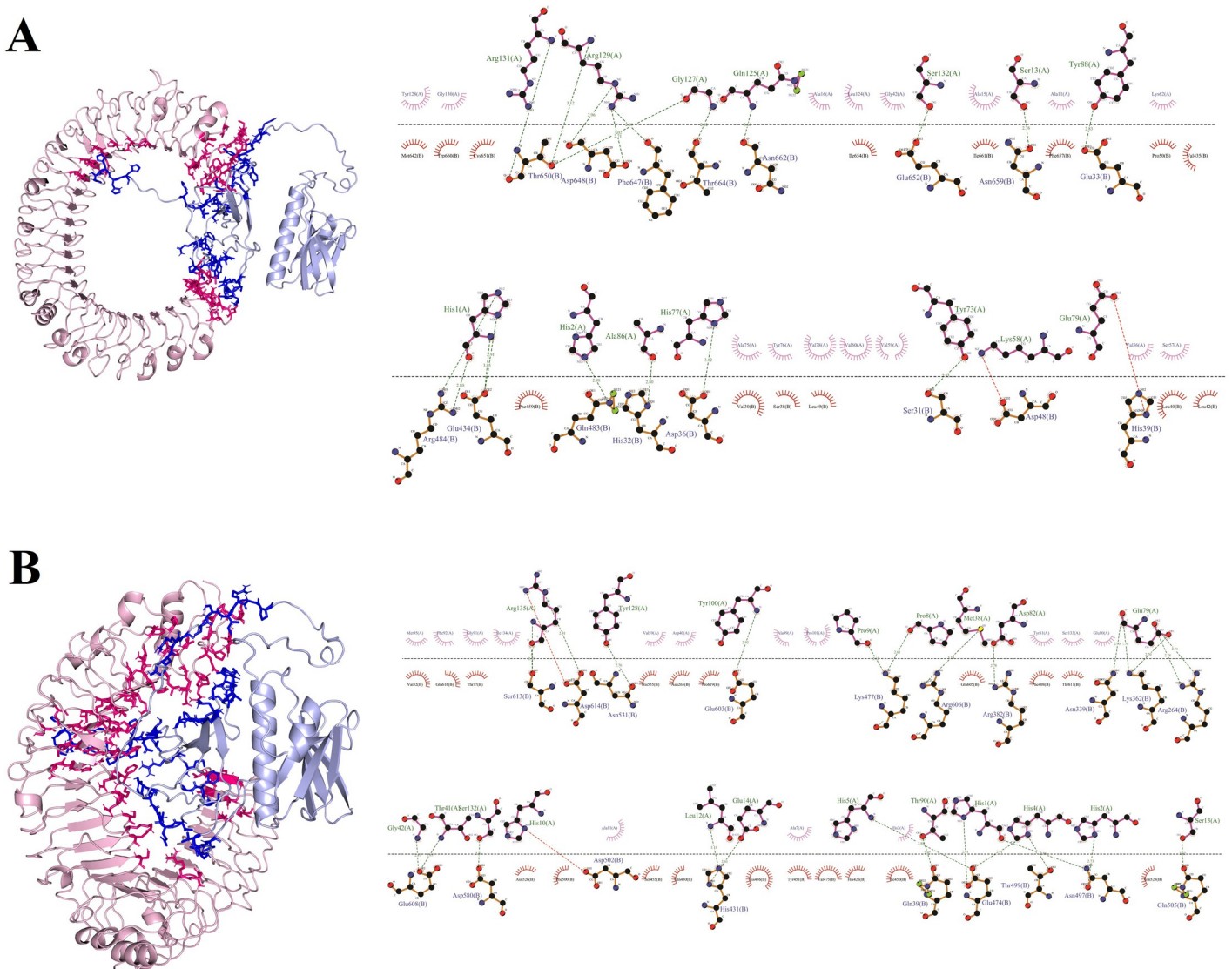

**Fig 5. Molecular docking between multiepitope vaccine candidate and TLR2 and TLR4 receptors using ClusPro 2.0.** Depiction of the docked conformation of A) multiepitope vaccine-TLR2 and B) multiepitope vaccine-TLR2 and representation of the interactions in complex using LigPlot. The multiepitope vaccine is shown in light blue and the TLR2 andTLR4 are shown in light pink.

as indicated by the high Rg descriptor. The vaccine-TLR2 and vaccine-TLR4 complexes had an estimated mean Rg value of 35.69 Å and 32.37 Å, respectively. Similar to the RMSD results, the Rg plots of these complexes remain steady throughout the simulation, indicating that the tertiary structure is finely compacted. Additionally, each complex was analyzed for solvent-accessible surface area (SASA), an important parameter for defining protein hydrophobicity. The mean SASA for the vaccine-TLR2 and vaccine TLR4 complexes was found to be 412.66 nm2 and 427.05 nm2, respectively. The SASA plot of the vaccine-TLRs complexes displayed a slightly decreasing trend, indicating that the hydrophobic core was exposed to less solvation as unfolding progressed, resulting in increased protein stability (Fig 6D). In addition, the number of hydrogen bonds produced or broken throughout the molecular simulation was ascertained by analyzing the average number of hydrogen bonds in each frame over time. The number of

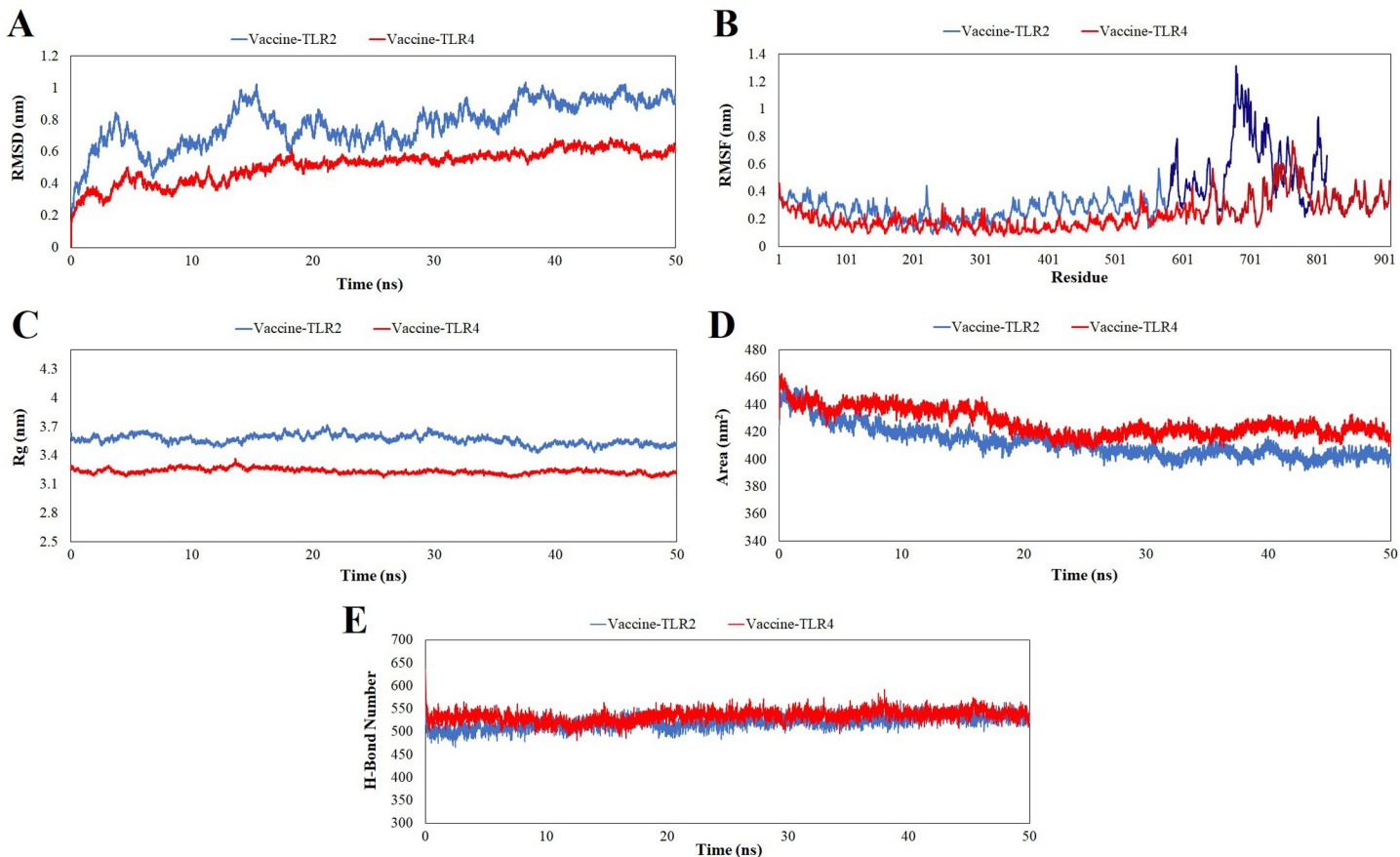

**Fig 6. MD simulation results. A**). RMSD curve for vaccine-TLRs complexes, **B**) RMSF curve for vaccine-TLRs complexes, **C**) Variation in the Radius of Gyration (Rg) of the complexes, D) Solvent-accessible surface area (SASA) of the complexes, E) Changes in the number of hydrogen bonds (H-bonds) in the complexes.

hydrogen bonds established in both systems remained almost constant across 50 ns of the molecular dynamic simulation, indicating the stability of the molecular structures (Fig 6E). Furthermore, the detailed energy analysis presented in Table 5 indicates that the multiepitope vaccine candidate exhibited a strong affinity for both TLR2 and TLR4 molecules. Specifically, the binding energy for the vaccine-TLR2 and vaccine-TLR4 complexes were -47.53 kcal/mol and -50.67 kcal/mol, respectively (Table 5). Overall, the results of all MM-PBSA calculations indicated that the multiepitope vaccine candidate can stably bind to the TLR2 and TLR4.

## Immune responses induced by the multiepitope vaccine candidate

Our study revealed heightened immunoglobulin activities, including IgM, IgM+IgG, IgG1 +IgG2, and IgG1 antibodies, as well as lowered antigen levels following the three vaccine candidate injections. Notably, the initial dose did not result in increased antibody levels, whereas the subsequent two doses did (Fig 7A). Additionally, after each injection, the levels of cytokines including interferon gamma (IFN-γ), interleukin-2 (IL-2), interleukin-10 (IL-10), and transforming growth factor beta (TGF-β) increased. After the first two injections, IFN-γ, IL-10, TGF-β, and levels increased but decreased after the third injection (Fig 7B). Following each dose, the total number of B cells increased, as did the number of IgM and IgG1 B cell isotypes. Additionally, with each injection, the population of active B cells and the number of B memory

**Table 5. The contribution of various energy components in the ΔGbind (kJ/mol) between multiepitope vaccine candidate and the TLR2 and TLR4.**

| Complex | VDW[a] (kcal/mol) | ELE[b] (kcal/mol) | GB[c] (kcal/mol) | SA[d] (kcal/mol) | Total[e] (kcal/mol) |
|---|---|---|---|---|---|
| Vaccine-TLR2 | -100.81 | -844.5 | 911.45 | -13.67 | -47.53 |
| Vaccine-TLR4 | -120.1 | -923.31 | 1008.55 | -15.82 | -50.68 |

[a] Van der waal energy

[b] Electrostatic energy

[c] Polar solvent energy

[d] Nonpolar solvent energy

[e] Binding free energy

cells increased, indicating the possibility of significant memory formation (Fig 7C and 7D). Comparable patterns were noted for cytotoxic and helper T cells. The populations of duplicating and resting HTL, as well as active HTL and CTL, were elevated and decreased slightly after the third injection. However, the resting HTC population decreased following the first injection, although it increased after the third injection (Fig 7E and 7F). In conclusion, our immune simulation predicted that multiepitope vaccine candidate injections would elicit a strong immune response.

## Codon optimization and *in-silico* cloning

Using the Java Codon Adaptation Tool (JCat), the multiepitope vaccine candidate underwent codon optimization to achieve the highest level of protein expression in E. coli (K12). The data demonstrated that the optimized multiepitope vaccine candidate consisted of 804 nucleotides with a CAI value of 1, indicating that the common codons were predominantly utilized. Moreover, the average GC content of the adapted sequence of the multiepitope vaccine candidate was 50.43%, which was within the optimal range (30–70%) inside the *E. coli* host. The sequence of the optimized multiepitope vaccine candidate was inserted between the *Nco*I and *Xho*I restriction sites of the pET28a (+) plasmid to create a recombinant plasmid (Fig 8).

## Discussion

Oncogenic human papillomaviruses are the leading cause of cervical cancer and other malignancies, including mouth, throat, gastric, penis, anus, vagina, and vulva cancer [61, 62]. The L1 protein in the viral capsid plays an important role in the infection process [10, 63]. Developing a vaccine based on L1 protein can potentially prevent HPV infection and related cancers [64]. However, current HPV vaccines do not provide complete protection against all HPV types [65], and their complex manufacturing processes, which involve insect or yeast cell expression systems, contribute to high costs. This high cost is a significant challenge for vaccination programs, particularly in developing countries where greater than 80% of cervical cancer cases occur [66–68]. Therefore, it is crucial to make concerted efforts to replace the existing vaccines with more affordable alternatives. Additionally, the efficacy of the candidate vaccine should be sufficiently high to generate robust cellular and humoral immune responses against major HPV types that cause cervical cancer and other HPV-related malignancies.

Reverse vaccinology is a promising approach for rapid development of peptide-based vaccines. This approach involves the use of immunodominant epitopes predicted in silico to accelerate vaccine development, achieve greater cost-effectiveness, enhance accuracy, and ensure vaccine safety. This method has gained significant attention in recent years because of its potential to overcome the limitations of traditional vaccine development processes and has been broadly employed to design novel vaccines against various pathogens such as

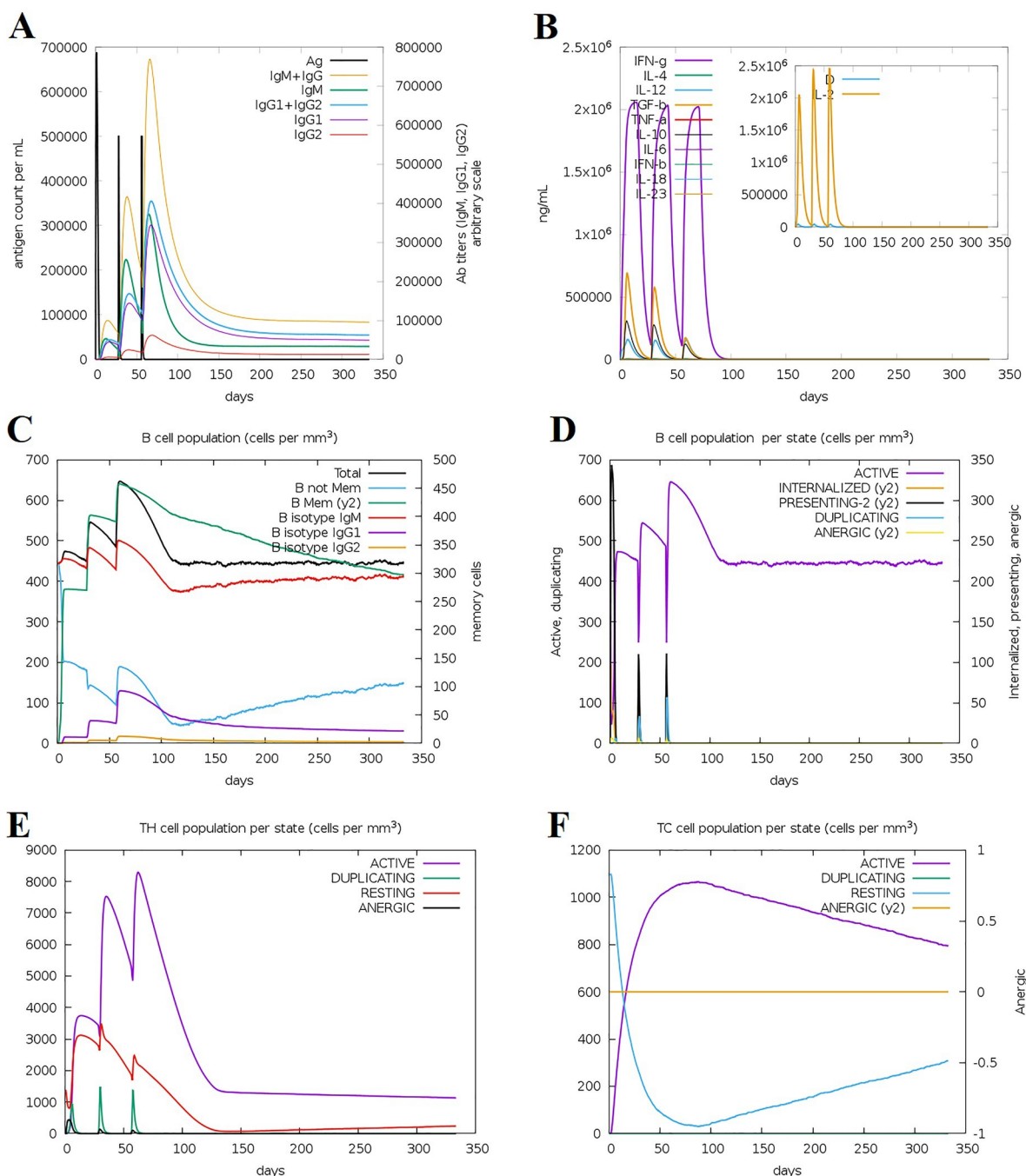

**Fig 7. Simulation of the immune response against the vaccine candidate using C-ImmSim server.** (A) Immunoglobulin production in response to immunization with the proposed multiepitope vaccine candidate. Specific subclasses are shown in different colors. (B) Levels of cytokines: The main diagram shows the cytokine levels after the multiepitope vaccine candidate injection. The inset plot shows the levels of the white blood cell growth factor IL-2 and the general activation signal of macrophage D. (C) Evolution of B cell populations after three vaccinations of the multiepitope vaccine candidate. (D) Production of B cells after the injection. Active B cells (depicted in purple) showed the highest secretion levels compared to other B cell subtypes. (E) Production of CD4 helper T cells in response to antigen exposure, including active, duplicating, resting, and anergic CD4 helper T cells. (F) Cytotoxic T cells production. The "resting state" indicates that no antigen occurs by cytotoxic T cells, while the "anergic state" indicates T cell tolerance to antigens due to repeated exposure.

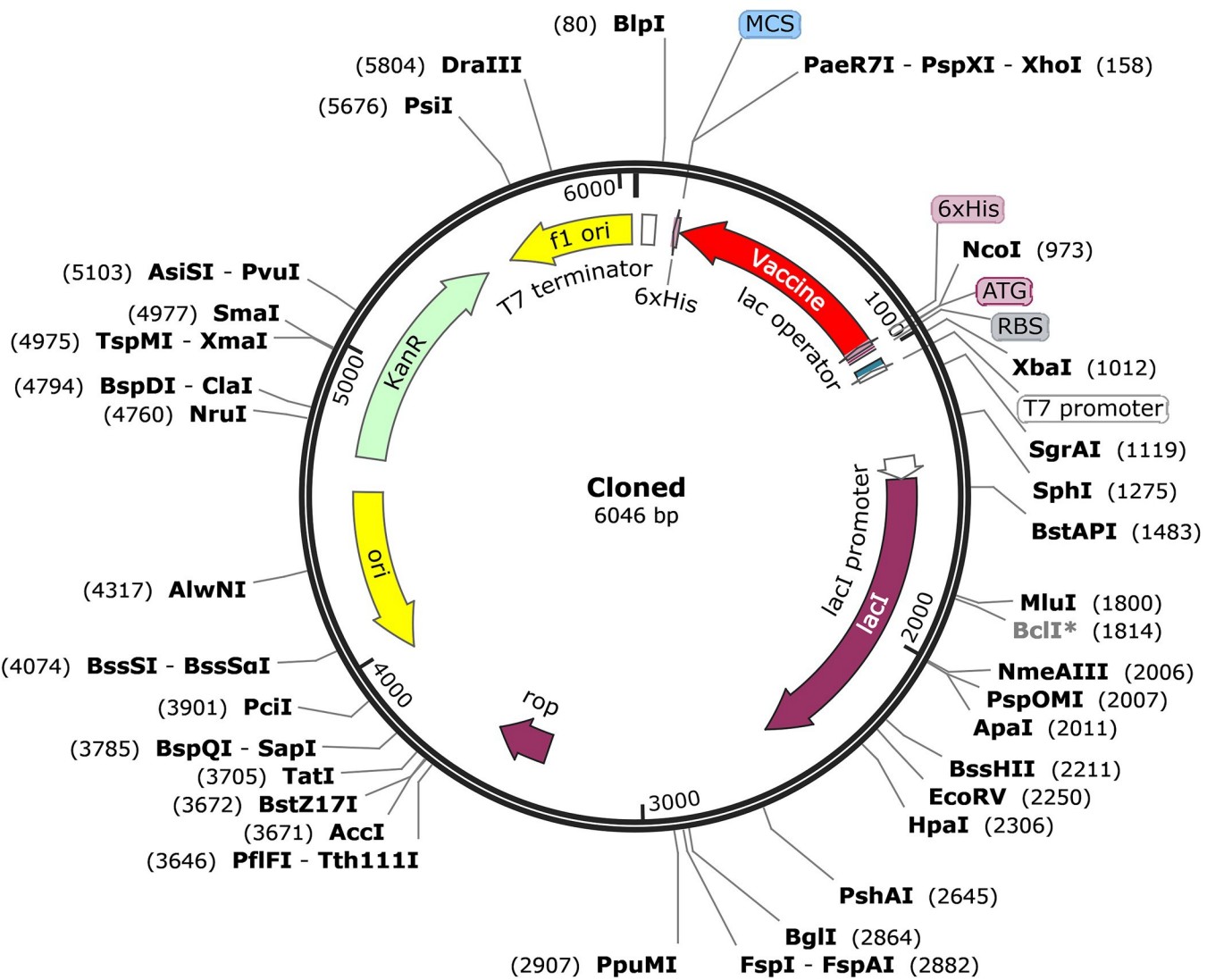

**Fig 8. *In silico* cloning of designed multiepitope vaccine construct.** The optimized DNA sequence of the designed multiepitope vaccine construct (shown in red) was cloned between the *Nco*I and *Xho*I restriction enzyme sites into the expression vector pET-28a (+).

*Mycobacterium Tuberculosis* [69, 70], *Listeria monocytogenes* [71], *Acinetobacter baumannii* [72], *Enterococcus faecium* [73], *Klebsiella pneumoniae* [74], *Pseudomonas aeruginosa* [75], *Leishmania donovani* [76], *Salmonella* [77], *Staphylococcus aureus* [78], influenza A virus [79, 80], SARS-CoV-2 [81–83], monkeypox virus [39, 84], and HPV16/18 [85].

The primary objective of this study was to develop a highly effective broad-spectrum HPV vaccine targeting immunodominant B-cell linear and T-cell epitopes derived from the most prevalent low- and high-risk HPV types. HPV L1 capsid protein is a favored target for vaccine development because of its high level of conservation among HPV types and the presence of numerous recognized neutralizing epitopes [86]. In the present investigation, a multiepitope vaccine candidate was developed using immunodominant B-cell and T-cell epitopes derived from various low- and high-risk types of human papillomavirus (HPV). Two protein adjuvants, CTB and RS09, were incorporated in the final vaccine formulation.

A successful vaccine must elicit a strong T-cell and B-cell immune response; therefore, in these studies, bioinformatic resources were employed to predict and select epitopes from experimentally validated epitopes contained in the IEDB. As B cells are crucial in the generation of memory [87], we screened linear B-cell epitopes present in the HPV L1 protein. Furthermore, the immune response driven by T cells triggered by CTL and HTL can induce a strong immune response against pathogens [88]. Given the importance of T-cell epitope-based vaccination, we considered HTL and CTL epitopes that could bind diverse HLA-A and HLA-B alleles with high affinity to construct a multiepitope vaccine. We used IEDB MHC-I and MHC-II binding predictions to predict CTL epitopes for the 27-allele HLA reference set class I alleles [89] and HTL epitopes for the 7-allele HLA reference set, which is known for its high population frequency [90]. IFN-γ blocks the expression of HPV mRNA in cervical cancer cells [91, 92]. Therefore, we identified IFN-γ-inducing epitopes within selected MHC II epitopes and incorporated them into the candidate vaccine (Table 1). Finally, we selected the highly immunogenic, non-toxic, and non-allergenic LBL, CTL, and HTL epitopes. Given the population coverage, our selected epitopes were expected to protect 99.26% of the global population with an average hit rate of 8.53 (Fig 2). To develop a universal broad-spectrum multiepitope HPV vaccine, it is necessary to perform conservation analysis of the selected epitopes. The sequence similarity of the selected epitopes ranged from 70% to 100% among the 17 examined HPV subtypes (S2-S4 Tables in S1 File), allowing for more effective formulations and broader immunity.

Tumor development is commonly linked to abnormal TLRs expression and chronic inflammation; however, the precise mechanism remains unclear. TLR agonists are widely employed as vaccine adjuvants [93]. Furthermore, studies have shown that a combining two or three TLR agonists can significantly increase both humoral and cellular immune responses [94–96]. The use of a suitable adjuvant in vaccine production can improve the efficacy of target antigens, increase the stability and potency of the vaccine in those who do not respond well, and produce stronger and longer-lasting immune responses. Based on Matos et al. (2017), TLR2 and TLR4 expression levels were found to be higher in cervical cancer and precancerous lesions than in normal controls [97]. CTB is a non-toxic subunit of cholera toxin and is classified as a pathogen-associated molecular pattern (PAMPs) or Toll-like receptor (TLR) ligand. Binding of the antigen to CTB enhances receptor-mediated uptake and subsequent antigen presentation by antigen-presenting cells (APCs) [98]. The heat-labile enterotoxin B subunit (LTB) is a protein that binds TLR2. Heat-labile enterotoxin B (LTB) is a protein that binds TLR2. Therefore, TLR2 was considered as the receptor for CTB in our study because of the high degree of sequence and structural similarity between CTB and LTB [99, 100]. Similarly, to improve antigenicity and immunogenicity, the synthetic TLR4 agonist RS09 (sequence: APPHALS) was added to the N-terminus of the multiepitope vaccine candidate as an adjuvant [101].

The antigenicity, allergenicity, toxicity, solubility, and physicochemical properties of the candidate vaccine were evaluated. The proposed vaccine construct was antigenic, non-toxic, non-allergenic, and soluble, suggesting its efficacy in stimulating strong immune responses, without causing harmful allergic reactions. The physicochemical characteristics indicate that this multiepitope vaccine candidate is suitable because of its low molecular weight (less than 100 kDa) [102], excellent thermal stability, and strong interaction with water molecules. Furthermore, the secondary structure of this multiepitope vaccine candidate showed a notable coil structure, which is critical for improving protein flexibility and antibody-binding ability. The 3D structure of the multiepitope vaccine was generated using the AlphaFold2 server and refined using the GalaxyRefine server, resulting in improved overall quality of the 3D structure of the vaccine. We conducted in silico analysis using the ElliPro server to predict continuous

B-cell epitopes. In silico analysis revealed 12 discontinuous epitopes, comprising 137 residues in the multiepitope vaccine construct.

The expression of TLR2 and TLR4 immune receptors is higher in human cervical cancer than that of other TLRs. Therefore, a molecular docking analysis was conducted to evaluate the interactions between the vaccine and TLR2 and TLR4. The constructed vaccine displayed hydrogen and hydrophobic interactions with good binding energies upon interaction with both the TLRs. The stability of the vaccine construct and TLR2 and TLR4 complexes was validated using MD simulations. These results indicated stable molecular interactions between the vaccine and immunological receptors, thereby ensuring the molecular stability of the multiepitope vaccine complex in a cellular environment. Immune stimulation indicated that the multiple-epitope vaccine candidate elicits a complex immune response. Immunoglobulin activities, such as IgM, IgM + IgG, IgG1, and IgG1 + IgG2 antibodies, dramatically increased after the three vaccine injections. Cytokine levels, including TGF-β, IFN-γ, IL-2, and IL-10, increased after each vaccination; however, they decreased after the third dose. T and B cells also exhibit dynamic shifts with the potential for robust memory formation and higher numbers of active B and T cell populations. According to these results, injections of multiple-epitope vaccines may coordinate diverse immune responses, offering important new information for immunization plans in the future.

In silico molecular cloning was performed to prepare this construct for downstream processing. Codon adaptation was performed to achieve high expression levels in *E. coli*. GC content (50.43%) and CAI score (1) were favorable for high expression. Bioinformatics analyses of this vaccine candidate are promising.

Multiepitope-based vaccines have various advantages over traditional vaccines, including safety, high potency, stability, and the ability to target the immune response to specific antigenic epitopes. Immunoinformatics can be used to develop a broad-spectrum HPV vaccine and analyze it using immune simulation, which is a more time-efficient and cost-effective method than direct in vitro and in vivo experiments. Targeting highly conserved epitopes has been the focus of efforts to develop universal vaccines. However, in several studies, distinct epitopes from various HPV types have been combined to produce cross-reactive protection against various HPV types [103, 104], which it is necessary to address the lack of broad-spectrum cross-protection and its poor immunogenicity. Our study established and validated a multiepitope vaccine in silico with cross-protective immunity against various HPV types. Nevertheless, additional wet laboratory animal experiments are required to assess the effectiveness and safety of the designed multiepitope vaccine candidate, which will be conducted in our subsequent studies.

## Conclusion

When the development of an effective vaccine against multiple viral variants is targeted, immunoinformatics has great potential for the design of multiepitope vaccines. In this study, we employed various immunoinformatic tools to develop a novel broad-spectrum multiepitope vaccine against human papillomavirus (HPV). Our results indicate that the multiepitope vaccine candidate demonstrated promising antigenicity and immunogenicity, was non-allergenic and non-toxic, and could elicit robust immune responses without causing adverse effects. The interaction between the multiepitope vaccine and TLR2 and TLR4 was assessed using molecular docking and molecular dynamics simulations, and in silico cloning was performed to ensure that the vaccine could be expressed in significant amounts in *E. coli*.

In conclusion, our study suggests that the designed broad-spectrum multiepitope vaccine has the potential to provide cross-protective immunity against various HPV types, which

represents a significant step towards preventing HPV infections and associated cancers. Further animal experiments with this multiepitope vaccine may provide insights into its effectiveness and cross-protection against HPVs.

## Supporting information

**S1 File. Supplementary material is available on the publisher's website along with the published article.**
(DOCX)

## Author Contributions

**Conceptualization:** Maryam Ehsasatvatan, Bahram Baghban Kohnehrouz.

**Data curation:** Maryam Ehsasatvatan.

**Formal analysis:** Maryam Ehsasatvatan.

**Investigation:** Maryam Ehsasatvatan.

**Methodology:** Maryam Ehsasatvatan.

**Project administration:** Bahram Baghban Kohnehrouz.

**Software:** Maryam Ehsasatvatan.

**Supervision:** Bahram Baghban Kohnehrouz.

**Validation:** Maryam Ehsasatvatan, Bahram Baghban Kohnehrouz.

**Visualization:** Maryam Ehsasatvatan.

**Writing – original draft:** Maryam Ehsasatvatan.

**Writing – review & editing:** Bahram Baghban Kohnehrouz.

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
