## [Decision Letter · Decision Letter 0]

13 Aug 2024

PONE-D-24-29409Designing and Immunomolecular Analysis of a New Broad-Spectrum Multiepitope Vaccine Against Divergent Human Papillomavirus TypesPLOS ONE

Dear Dr. Baghban Kohnehrouz,

Thank you for submitting your manuscript to PLOS ONE. After careful consideration, we feel that it has merit but does not fully meet PLOS ONE’s publication criteria as it currently stands. Therefore, we invite you to submit a revised version of the manuscript that addresses the points raised during the review process.

We look forward to receiving your revised manuscript.

Kind regards,

Sheikh Arslan Sehgal, PhD

Academic Editor

PLOS ONE

Journal Requirements:

4. Please note that funding information should not appear in any section or other areas of your manuscript. We will only publish funding information present in the Funding Statement section of the online submission form. Please remove any funding-related text from the manuscript.

5. In the online submission form, you indicated that "The data that support the findings of this study are available from the corresponding author upon reasonable request"

Reviewers' comments:

Reviewer's Responses to Questions

**Comments to the Author**

1. Is the manuscript technically sound, and do the data support the conclusions?

Reviewer #1: Yes

Reviewer #2: No

2. Has the statistical analysis been performed appropriately and rigorously? 

Reviewer #1: N/A

Reviewer #2: N/A

3. Have the authors made all data underlying the findings in their manuscript fully available?

Reviewer #1: Yes

Reviewer #2: Yes

4. Is the manuscript presented in an intelligible fashion and written in standard English?

Reviewer #1: Yes

Reviewer #2: No

5. Review Comments to the Author

Reviewer #1: The manuscript entitled "Designing and Immunomolecular Analysis of a New Broad-Spectrum Multiepitope Vaccine Against Divergent Human Papillomavirus Types" reports the results of an immunoinformatics approach to develop a vaccine for the HPV virus. The computational details, results and discussion part is sound. The manuscript can be accepted after incorporating the following changes in the revised version:

1. In the introduction, explain why a new vaccine is needed compared to the three already approved vaccines. Discuss the limitations of the current vaccines and the need for a more broadly protective and therapeutic vaccine.

2. The authors used a threshold of 0.4 for determining the antigenicity of viral epitopes for B cell and CTL epitopes. However, for HTL epitopes, they used a threshold value of 0.5. Explain the rationale behind using different thresholds for different types of epitopes. Provide references or justification for the chosen thresholds.

3. Modify RS09 (APPHALS) with RS09 (sequence: APPHALS)

4. Clarify how TLR2 and TLR4 were selected for this study. Refer to the paper by DOI: 10.3390/pathogens9040292 for guidance on the selection of relevant TLRs.

5. In the molecular docking analysis, explain how the best model was selected out of the 30 conformations generated. Provide the criteria used for selection, such as binding energy, cluster size, or other relevant parameters.

6. Perform an NCBI protein BLAST analysis to assess if the vaccine candidate is considered an autoimminogen in humans. This will help evaluate the potential for autoimmune reactions.

7. Discuss how this study differs from previous immunoinformatics studies on HPV vaccine design. Identify if the epitopes predicted in this study have been predicted before or if they are novel.

8. Replace low-resolution figures with high-quality images to ensure clarity and readability. Most of the Figures are of low resolution.

9. In Figure 6, distinguish the RMSF of the vaccine and TLRs using different colors or annotations. This will help readers easily identify the RMSF of each component.

10. In the molecular docking results section, mark the amino acids of the multiepitope vaccine differently compared to those of the TLRs. This will make it easier for readers to identify which amino acids belong to which part of the system.

Reviewer #2: Thank you for submitting your manuscript, "Designing and Immunomolecular Analysis of a New Broad-Spectrum Multiepitope Vaccine Against Divergent Human Papillomavirus Types " .I appreciate the effort you and your co-authors have put into this work.

After a thorough evaluation, I regret to inform you that I must recommend the rejection of this manuscript. While the topic is of significant interest and relevance, there are several critical issues that need to be addressed:

1. All the existing licensed vaccines such as (Gardasil, Gardasil-9, and Cervarix) are made on the basis of L1 protein and have an effective preventive effect. The authors should have done something more innovative and used another protein or at least one more protein In addition to L1, they targeted.

2. The manuscript lacks the necessary scientific rigor in several key areas. The methodologies used, particularly in the immunoinformatics analyses, are not sufficiently detailed to allow for reproducibility. Additionally, the validation of the computational predictions through experimental data is limited, which diminishes the strength of the conclusions drawn.

3. Fluctuations associated with vaccine-receptor complexes are greater than 3 Å and the complex has not reached sufficient stability.

4. The authors did not provide MM-PBSA calculations of the simulated trajectory.

5. There are concerns regarding the interpretation of the data. Some conclusions appear to be overreaching based on the presented data, and additional context or supporting information is needed to substantiate these claims.

6. PLOS authors have the option to publish the peer review history of their article (what does this mean?). If published, this will include your full peer review and any attached files.

Reviewer #1: **Yes: **Abdul Rajjak Shaikh

Reviewer #2: No

---

## [Author Response · Author response to Decision Letter 0]

28 Aug 2024

August 15, 2024

Journal of Plos One 

Dear Editor, Dr. Sheikh Arslan Sehgal

We appreciate you and the reviewers for your precious time spent reviewing our paper and providing valuable comments. These are your valuable and insightful comments that lead to possible improvements in the current version. The authors have carefully considered the comments and tried their best to address every one of them. We hope the manuscript, after careful revisions, meets your high standards. The authors welcome any further constructive comments. 

Below, we provide the point-by-point responses. Reviewer comments and modifications in the manuscript have been highlighted in yellow.

In addition, as this manuscript is an integral part of the research project approved by the Iran National Science Foundation, it is imperative to indicate the source of funding in the Acknowledgment or Funding section of the manuscript. It is necessary to receive a certificate of completion for the project from the institution providing financial support. Therefore, we kindly request you to add this funding source to our manuscript as "This work is based upon research funded by Iran National Science Foundation (INSF) under project No. 4023935. The funders had no role in study design, data collection and analysis, decision to publish, or preparation of the manuscript.".

Sincerely,

Bahram Baghban Kohnehrouz, 

PhD in Mol Biol and Biotechnology,

Faculty of Agriculture

University of Tabriz

Tabriz, Iran.

Reviewer #1: 

1. In the introduction, explain why a new vaccine is needed compared to the three already approved vaccines. Discuss the limitations of the current vaccines and the need for a more broadly protective and therapeutic vaccine.

Response: In the fifth paragraph of the "Introduction" section, we have presented an explanation of the limitations of current vaccines and the necessity for a more comprehensive protective vaccine.

“Since 2006, when the HPV vaccine was first approved in the USA, real-world data have consistently demonstrated its safety and effectiveness in preventing and treating HPV infections and its associated diseases. However, various barriers have been identified, such as high vaccine costs, inaccessibility, and inadequate storage and transportation facilities (14). Moreover, in many low- and middle-income countries, public awareness regarding HPV-related disorders and national vaccination programs is lacking (15). The main challenge in implementing HPV vaccines is that they do not provide protection against all HPV types (16). To address these limitations, the next generation of HPV vaccines should focus on reducing the side effects associated with current vaccines using alternative adjuvants or other vaccine designs and high-valent vaccines based on recombinant vectors with a broad protection spectrum that can be administered by inhalation or the oral route. This is a major step in the treatment of cervical cancer.”

2. The authors used a threshold of 0.4 for determining the antigenicity of viral epitopes for B cell and CTL epitopes. However, for HTL epitopes, they used a threshold value of 0.5. Explain the rationale behind using different thresholds for different types of epitopes. Provide references or justification for the chosen thresholds.

Response: We utilized Vaxijen’s default threshold of 0.4 for viral models in order to have a greater number of antigen epitopes to choose from B cell epitopes and CTL epitopes. For HTL epitopes, a higher threshold was employed to accurately predict the antigenicity of the epitopes because of the larger number of predicted epitopes. However, it is important to note that all epitopes selected for vaccine design, as shown in Tables 1 and 2, had antigenicity values higher than 0.5.

.

3. Modify RS09 (APPHALS) with RS09 (sequence: APPHALS)

Response: Thank you for pointing this out. The change was made and highlighted in yellow.

4. Clarify how TLR2 and TLR4 were selected for this study. Refer to the paper by DOI: 10.3390/pathogens9040292 for guidance on the selection of relevant TLRs.

Response: In the fifth paragraph of the “Discussion” section, we have provided an explanation for our selection of TLR2 and TLR4 for molecular docking analysis in this present research.

“Tumor development is commonly linked to abnormal TLRs expression and chronic inflammation; however, the precise mechanism remains unclear. TLR agonists are widely employed as vaccine adjuvants (92). Furthermore, studies have shown that a combining two or three TLR agonists can significantly increase both humoral and cellular immune responses (93-95). The use of a suitable adjuvant in vaccine production can improve the efficacy of target antigens, increase the stability and potency of the vaccine in those who do not respond well, and produce stronger and longer-lasting immune responses. Based on Matos et al. (2017), TLR2 and TLR4 expression levels were found to be higher in cervical cancer and precancerous lesions than in normal controls (96). CTB is a non-toxic subunit of cholera toxin and is classified as a pathogen-associated molecular pattern (PAMPs) or Toll-like receptor (TLR) ligand. Binding of the antigen to CTB enhances receptor-mediated uptake and subsequent antigen presentation by antigen-presenting cells (APCs) (97). The heat-labile enterotoxin B subunit (LTB) is a protein that binds TLR2. Heat-labile enterotoxin B (LTB) is a protein that binds TLR2. Therefore, TLR2 was considered as the receptor for CTB in our study because of the high degree of sequence and structural similarity between CTB and LTB (98, 99). Similarly, to improve antigenicity and immunogenicity, the synthetic TLR4 agonist RS09 (sequence: APPHALS) was added to the N-terminus of the multiepitope vaccine candidate as an adjuvant (100).”

5. In the molecular docking analysis, explain how the best model was selected out of the 30 conformations generated. Provide the criteria used for selection, such as binding energy, cluster size, or other relevant parameters.

Response: In the “Methods” section and the “Molecular docking” subsection, we explained that “Docking was performed without the use of predetermined active site information, and the centers of the largest clusters were used as plausible complex models of low-energy docked structures”. However, in the opinion of the respected referee, more explanations are provided in the “Results” section.

6. Perform an NCBI protein BLAST analysis to assess if the vaccine candidate is considered an autoimminogen in humans. This will help evaluate the potential for autoimmune reactions.

Response: Thank you for pointing out this shortage. The analysis has been conducted for the chosen epitopes, however, we have implemented the analysis for the vaccine design, as recommended by the reviewer, and the results have been incorporated into the revised manuscript. The alterations are indicated in yellow.

7. Discuss how this study differs from previous immunoinformatics studies on HPV vaccine design. Identify if the epitopes predicted in this study have been predicted before or if they are novel.

Response: Previous scholarly research frequently employed two or more proteins for selecting epitopes and designing vaccines. According to prior investigations, the L2 protein is less effective in stimulating the immune system than the L1 protein. E-proteins have also been utilized in the development of therapeutic vaccines. Our study is the only one to employ the L1 protein for epitope prediction and vaccine design. Moreover, the epitopes selected in our study were novel and have not been utilized in previous studies for vaccine design.

8. Replace low-resolution figures with high-quality images to ensure clarity and readability. Most of the Figures are of low resolution.

Response: We have replaced all figures with high-resolution figures.

9. In Figure 6, distinguish the RMSF of the vaccine and TLRs using different colors or annotations. This will help readers easily identify the RMSF of each component. 

Response: We thank the reviewer for pointing this out. We distinguished the RMSF of the vaccine and TLRs using different colors.

10. In the molecular docking results section, mark the amino acids of the multiepitope vaccine differently compared to those of the TLRs. This will make it easier for readers to identify which amino acids belong to which part of the system.

Response: Thank you for pointing out this. We updated the manuscript in this regard, and the changes were highlighted. 

Reviewer #2: 

1. All the existing licensed vaccines such as (Gardasil, Gardasil-9, and Cervarix) are made on the basis of L1 protein and have an effective preventive effect. The authors should have done something more innovative and used another protein or at least one more protein In addition to L1, they targeted.

Response: HPVs are small, non-enveloped, double-stranded DNA viruses with genomes comprising eight ORFs organized into three functional regions: early (E), late (L), and non-coding long control regions (LCR). The E region genes code for proteins E1, E2, E4, E5, E6, E7, and E8, which are crucial for viral replication and pathogenicity. The L region genes encode L1 and L2 capsid proteins, which are vital for virion assembly. LCR genes play a crucial role in viral DNA replication and transcription, and exhibit tropism in epithelial cells. The L1 protein, a major capsid component, includes constant and variable regions that contribute to surface antigenicity and interacts with host plasma membrane receptors before penetration. It also serves as a target for host antibody generation, and is vital for the diversity of HPV genotypes. All currently licensed vaccines use the full-length L1 protein for VLP construction. They do not provide complete protection against all HPV types and their complex manufacturing processes, which involve insect or yeast cell expression systems, contribute to high costs. Cervarix contains HPV16 and 18 VLPs, while Gardasil includes VLPs against HPV6, 11, 16, and 18. Gardasil 9 contains five additional VLPs against HPV31, 33, 45, 52, and 58. Previous immunoinformatic studies have used the L2 protein to predict antigen epitopes for vaccine design; however, it did not stimulate the immune system as effectively as the L1 protein. E-proteins have also been employed in the development of therapeutic vaccines. The objective of our study was to design and create a multi-epitope vaccine that targets the most prevalent low- and high-risk human types, including five low-risk HPV types (HPV6, 11, 42, 43, and 44) and 12 high-risk HPV types (HPV16, 18, 31, 33, 35, 39, 45, 51, 52, 56, 58, and 59), using the major capsid protein L1 to achieve higher cost-effectiveness in vaccine development.

2. The manuscript lacks the necessary scientific rigor in several key areas. The methodologies used, particularly in the immunoinformatics analyses, are not sufficiently detailed to allow for reproducibility. Additionally, the validation of the computational predictions through experimental data is limited, which diminishes the strength of the conclusions drawn.

Response: The present manuscript details the outcome of an immunoinformatics investigation aimed at devising and proposing an efficacious vaccine against the most prevalent types of HPV. The study was designed and carried out using standard techniques in immunoinformatics and bioinformatics, and all reported findings were derived from dependable software and servers commonly employed in such analyses. This method has been broadly employed to design novel vaccines against various pathogens. However, as indicated by the referee, computational predictions should be verified through experimental testing. Thus, as articulated in the Conclusion section, the results of this study warrant further investigation in animal models, which is the focus of our ongoing research.

3. Fluctuations associated with vaccine-receptor complexes are greater than 3 Å and the complex has not reached sufficient stability.

Response: The reason for the higher root-mean-square deviation (RMSD) was the complexity of the system. It is not unusual to observe high RMSD values in systems with flexible or large components, complex protein-ligand interactions, or significant conformational changes. However, it is important to note that a high RMSD does not necessarily indicate a problem. Depending on the complexity of the system and the goals of the simulation, a high but stable RMSD is acceptable.

4. The authors did not provide MM-PBSA calculations of the simulated trajectory.

Response: We used the PRODIGY server to determine the binding free energy of our simulated trajectory. However, in accordance with the suggestions of the esteemed reviewer, we have incorporated the MM-PBSA calculations into the manuscript and distinguished these modifications by highlighting them in yellow.

5. There are concerns regarding the interpretation of the data. Some conclusions appear to be overreaching based on the presented data, and additional context or supporting information is needed to substantiate these claims.

Response: The findings of this study were derived from precise bioinformatics software and servers and have been meticulously interpreted by the authors. As mentioned in the Conclusion section, the exact verification of the data obtained from this study will be provided in future experimental studies. However, if the esteemed reviewer offers more in-depth opinions regarding the results that necessitate greater precision or interpretation, we will attempt to rectify them.

---

## [Decision Letter · Decision Letter 1]

18 Sep 2024

Designing and Immunomolecular Analysis of a New Broad-Spectrum Multiepitope Vaccine Against Divergent Human Papillomavirus Types

PONE-D-24-29409R1

Dear Dr. Baghban Kohnehrouz,

We’re pleased to inform you that your manuscript has been judged scientifically suitable for publication and will be formally accepted for publication once it meets all outstanding technical requirements.

Kind regards,

Sheikh Arslan Sehgal, PhD

Academic Editor

PLOS ONE

Additional Editor Comments (optional):

Reviewers' comments:

Reviewer's Responses to Questions

**Comments to the Author**

1. If the authors have adequately addressed your comments raised in a previous round of review and you feel that this manuscript is now acceptable for publication, you may indicate that here to bypass the “Comments to the Author” section, enter your conflict of interest statement in the “Confidential to Editor” section, and submit your "Accept" recommendation.

Reviewer #1: All comments have been addressed

Reviewer #2: All comments have been addressed

2. Is the manuscript technically sound, and do the data support the conclusions?

Reviewer #1: Yes

Reviewer #2: Yes

3. Has the statistical analysis been performed appropriately and rigorously? 

Reviewer #1: N/A

Reviewer #2: N/A

4. Have the authors made all data underlying the findings in their manuscript fully available?

Reviewer #1: Yes

Reviewer #2: Yes

5. Is the manuscript presented in an intelligible fashion and written in standard English?

Reviewer #1: Yes

Reviewer #2: Yes

6. Review Comments to the Author

Reviewer #1: All of my concerns have been addressed, and I am satisfied with the current version. Now it can be accepted in present form.

Reviewer #2: I am grateful to the authors for carefully considering the comment of reviewer. I hope this work will have a good effect in dealing with cervical cancer.

7. PLOS authors have the option to publish the peer review history of their article (what does this mean?). If published, this will include your full peer review and any attached files.

Reviewer #1: **Yes: **Abdul Rajjak Shaikh

Reviewer #2: No

---

## [Editor Report · Acceptance letter]

24 Sep 2024

PONE-D-24-29409R1 

PLOS ONE

Dear Dr. Baghban Kohnehrouz, 

I'm pleased to inform you that your manuscript has been deemed suitable for publication in PLOS ONE. Congratulations! Your manuscript is now being handed over to our production team.

Kind regards, 

on behalf of

Dr Sheikh Arslan Sehgal 

Academic Editor

PLOS ONE